# Distinct pulmonary and systemic effects of dexamethasone in severe COVID-19

Lucile P. A. Neyton [1,33], Ravi K. Patel [2,33], Aartik Sarma [1,33], UCSF COMET Consortium*, Andrew Willmore[1], Sidney C. Haller[1], Kirsten N. Kangelaris[3], Walter L. Eckalbar[1,2], David J. Erle [1,2,4,5], Matthew F. Krummel [6], Carolyn M. Hendrickson [1], Prescott G. Woodruff[1], Charles R. Langelier [7,8], Carolyn S. Calfee[1,4,9] & Gabriela K. Fragiadakis [2,10] ✉

Dexamethasone is the standard of care for critically ill patients with COVID-19, but the mechanisms by which it decreases mortality and its immunological effects in this setting are not understood. Here we perform bulk and single-cell RNA sequencing of samples from the lower respiratory tract and blood, and assess plasma cytokine profiling to study the effects of dexamethasone on both systemic and pulmonary immune cell compartments. In blood samples, dexamethasone is associated with decreased expression of genes associated with T cell activation, including *TNFSFR4* and *IL21R*. We also identify decreased expression of several immune pathways, including major histocompatibility complex-II signaling, selectin P ligand signaling, and T cell recruitment by intercellular adhesion molecule and integrin activation, suggesting these are potential mechanisms of the therapeutic benefit of steroids in COVID-19. We identify additional compartment- and cell- specific differences in the effect of dexamethasone that are reproducible in publicly available datasets, including steroid-resistant interferon pathway expression in the respiratory tract, which may be additional therapeutic targets. In summary, we demonstrate compartment-specific effects of dexamethasone in critically ill COVID-19 patients, providing mechanistic insights with potential therapeutic relevance. Our results highlight the importance of studying compartmentalized inflammation in critically ill patients.

Moderate doses of corticosteroids, including dexamethasone, decrease mortality in patients with severe COVID-19 in clinical trials[1]. Conversely, steroids may increase mortality in COVID-19 patients without hypoxemia[2], and higher doses of dexamethasone may increase mortality in hypoxemic, non-ventilated patients[3]. While randomized controlled trials of steroids in patients with COVID-19 have transformed clinical practice, the cell- and compartment-specific effects of corticosteroids in these patients are not well understood.

[1]Division of Pulmonary, Critical Care, Allergy and Sleep Medicine, University of California, San Francisco, CA, USA. [2]UCSF CoLabs, University of California San Francisco, San Francisco, CA, USA. [3]Division of Hospital Medicine, University of California, San Francisco, CA, USA. [4]Department of Medicine, University of California, San Francisco, CA, USA. [5]Lung Biology Center, University of California, San Francisco, CA, USA. [6]Department of Pathology, University of California, San Francisco, CA, USA. [7]Chan Zuckerberg Biohub, San Francisco, CA, USA. [8]Division of Infectious Diseases, University of California, San Francisco, CA, USA. [9]Department of Anesthesia, University of California, San Francisco, CA, USA. [10]Division of Rheumatology, University of California, San Francisco, CA, USA. [33]These authors contributed equally: Lucile P. A. Neyton, Ravi K. Patel, Aartik Sarma. *A list of authors and their affiliations appears at the end of the paper. ✉ e-mail: gabriela.fragiadakis@ucsf.edu

Dexamethasone is classically considered a non-specific and potent systemic anti-inflammatory medication, but it has pleiotropic effects on inflammatory signaling, wound healing, and metabolism in experimental models[4]. In experimental studies in animal models and human volunteers, dexamethasone and other corticosteroids have distinct effects on systemic versus pulmonary inflammation[5], and several studies have identified cell-specific effects of glucocorticoids[6]. While a small number of studies have described the effects of corticosteroids on blood and lung gene expression in COVID-19[7,8], no work has yet comprehensively evaluated effects across gene, protein, and cellular levels in both systemic circulation and respiratory tract. Further understanding the cell- and compartment-specific effects of dexamethasone in severe COVID-19 may elucidate the therapeutic effects of steroids in these patients and further our understanding of the role of steroids in other viral infections and/or the acute respiratory distress syndrome (ARDS) more generally.

Here, we use single-cell RNA sequencing to study peripheral blood and tracheal aspirate (TA) from a multi-center observational cohort of patients with COVID-19 before and after dexamethasone became standard of care, using data generated as part of the COMET and IMPACC studies[9,10]. We integrate this data with cytokine and gene expression data from blood and compare it to two publicly available datasets. We identify several cell-specific differences in the pulmonary and systemic effects of dexamethasone in mechanically ventilated patients with COVID-19 ARDS, many of which were reproducible in the external datasets. Through receptor-ligand analysis, we also detect signatures of injury resolution and reduced antigen presentation and T cell recruitment in dexamethasone-treated patients, returning to levels observed in healthy controls. This work highlights the importance of studying both local and systemic inflammatory signaling in acute respiratory disease and identifying biological pathways that may represent future therapeutic targets.

## Results

We conducted a prospective case-control study of mechanically ventilated adults (age ≥ 18) with COVID-19 acute respiratory distress syndrome (ARDS) at two academic hospitals: the University of California, San Francisco Medical Center (UCSFMC), and the Zuckerberg San Francisco General Hospital (ZSFG). Patients were enrolled into an observational cohort starting in April 2020. At both sites, patients did not routinely receive corticosteroids for COVID-19 ARDS prior to the publication of the RECOVERY trial in July 2020, at which time dexamethasone was promptly introduced as a treatment for patients hospitalized with severe COVID-19. We studied patients enrolled before and after this rapid change in the standard of care, which enabled a multi-omic characterization of the effects of dexamethasone in patients with COVID-19 ARDS.

For this study, we included patients admitted to the ICU with at least one biospecimen (TA, blood, or plasma) collected (Fig. 1a) while they were mechanically ventilated. We excluded patients who received steroids for an indication other than COVID-19 and those who received other immunosuppressive drugs (e.g., tocilizumab, baricitinib), leaving a final sample size of 27 patients who received at least one dose of 6 mg dexamethasone at the time of initial biosampling (Dex) and 16 patients who did not receive dexamethasone (NoDex) prior to specimen collection (Fig. S1, Table S1). An overview of patients included in the different analyses is provided (Fig. 1b). All included patients were recruited between April 2020 and March 2021.

### Dexamethasone modulates cytokine and immune cell gene expression in blood samples from patients with severe COVID-19

We first profiled a panel of 18 plasma cytokines (Table S2) previously associated with COVID-19 and ARDS pathophysiology[11] in Dex ($N = 15$) and NoDex ($N = 23$) subjects at the time of study enrollment. After adjusting for multiple hypothesis testing, we observed significantly

lower plasma IL-6 and IFN-gamma in Dex patients compared to NoDex patients (Fig. 1c). Conversely, we observed significantly higher levels of IL-10, a cytokine that suppresses inflammatory responses[12], in Dex patients treated with dexamethasone (Fig. 1c). Other cytokines did not present significantly different levels across treatment groups (Fig. S2A). Examination of times between first dexamethasone dose and sample collection demonstrated that these changes in cytokine levels persisted for at least 24 h after starting steroid treatment (Fig. S2B).

We then compared peripheral blood gene expression between the Dex ($N = 10$) and NoDex ($N = 11$) groups and found 4,050 differentially expressed genes (20% of protein coding genes tested) after adjusting for age and sex assigned at birth (adjusted $p < 0.1$) (Fig. 1d). Immune genes such as *TNFRSF4*, involved in T cell co-stimulation, and *IL21R*, involved in T-/B- and NK-cell activation, as well as several genes involved in allergic responses (*MS4A2*, *PTGDR2*) were downregulated in Dex patients. Genes upregulated in the Dex patients included *ADAMTS2*, a procollagen N-endopeptidase upregulated by TGF-beta that has been reported to be upregulated by glucocorticoids[13], and *RLN3*, involved in the response to DNA damage and repair[14]. Gene set enrichment analysis (GSEA) of results of the differential gene expression analysis identified 21 significantly dysregulated pathways in the Reactome database (adjusted $p < 0.1$) (Fig. S3). The most enriched pathways in Dex patients included metabolic pathways such as tricarboxylic acid cycle and several mitochondria-associated pathways, defense against pathogens, and interferon signaling. Conversely, NoDex patients had gene expression signatures consistent with the enrichment of sensory perception pathways possibly linked to differences in leukocyte populations[15], and the activation of cell survival related pathways such as fibroblast growth factor receptor (FGFR)- and G-protein-coupled receptor (GPCR).

### Supervised integrative analysis of blood transcriptomic and plasma cytokine data identifies co-varying responses to dexamethasone

We next designed an integrative analysis examining the effect of dexamethasone on gene expression and protein concentrations in all patients with both data types available from the same blood sample ($N = 10$ Dex patients and $N = 11$ NoDex). We used DIABLO[16], an implementation of partial least squares discriminant analysis, to identify components ("variates") shared across modalities that stratify based on dexamethasone treatment with the goal of identifying coordinated changes across gene expression and protein concentrations vs. changes independently observed in unique data types. Variate 1 clearly separated Dex from NoDex patients (Fig. 2a). When examining the contributions to variate 1 from the cytokine data, Dex patients were separated based on lower IP-10, which is involved in interferon gamma signaling; lower levels of the inflammatory cytokines IL-6 and IL-18; lower ICAM-1, which is involved in inflammation and leukocyte recruitment; and lower Ang-2, a facilitator of angiogenesis and antagonist to Ang-1. Dex patients were conversely separated by higher Ang-1, and higher levels of protein C and IL-10, reflecting the attenuated proinflammatory cytokine signaling observed in the unimodal analysis (Fig. 2b).

Gene set enrichment analysis of the transcriptomic contributions to variate 1 unexpectedly demonstrated relative elevation of innate immune response and cytokine signaling pathways in Dex patients compared to the NoDex patients (Fig. 2c). Covariation highlighted by DIABLO exposed a decrease in the inflammatory response in circulating cytokines, and an increase in inflammatory responses in peripheral blood gene expression. Pathways involved in defense against pathogens, as well as interferon signaling, were found to be enriched in Dex patients, consistent with the analysis of peripheral blood gene expression. Additionally, gene expression variation represented by variate 1 was associated with alterations in transcriptional regulation and specifically, to epigenetic-related processes.

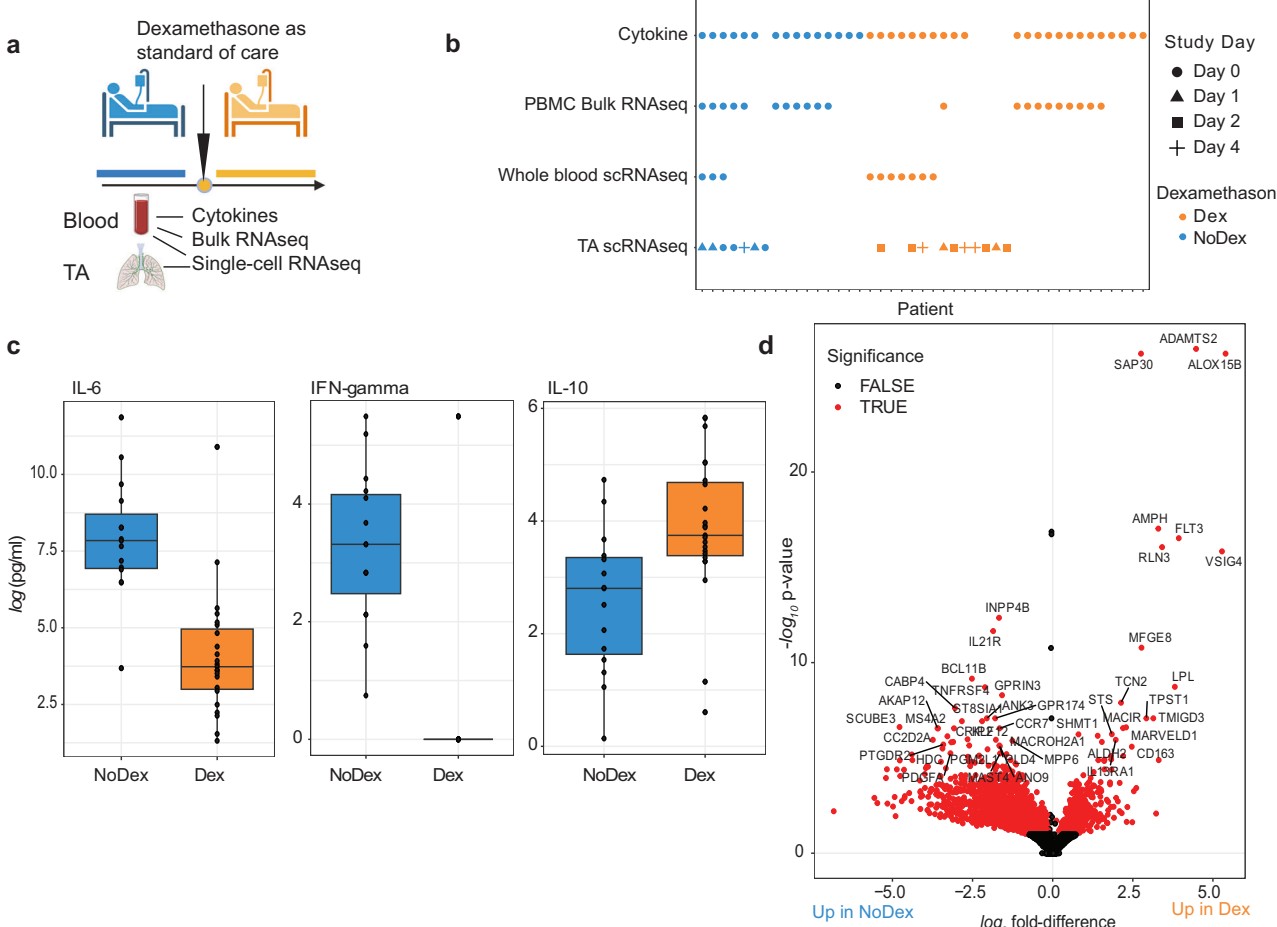

**Fig. 1 | Dexamethasone modulates cytokine and immune cell gene expression in the blood of patients with COVID-19. a** The introduction of dexamethasone (Dex) as standard of care for critically ill patients hospitalized with COVID-19 based on the results of the RECOVERY trial. Blood and tracheal aspirate (TA) samples were collected from intubated patients enrolled either before or after this change. Figure 1a Created with BioRender.com released under a Creative Commons Attribution-NonCommercial-NoDerivs 4.0 International license (https://creativecommons.org/licenses/by-nc-nd/4.0/deed.en). **b** Included patients and time points per analysis. A single sample was used per patient. Each patient was either treated with Dex (orange) or not (blue). Samples used in DIABLO analysis (Fig. 2) are the overlap in PBMC bulk RNA sequencing and plasma cytokine rows. **c** Individual plots of *log*-transformed significant cytokines IL-6, IL-10, and interferon gamma (IFN-gamma) (two-sided Wilcoxon rank-sum test, BH-adjusted $p < 0.1$). The median, first and third quartiles, and 1.5*interquartile range distance from the quartiles are represented using the center mark, hinges, and whiskers, respectively. $N = 23$ Dex, $N = 15$ NoDex. **d** Volcano plot of differential gene expression of PBMC RNA-seq data with DESeq2 (based on two-sided negative binomial generalized linear models). Significance determined using a BH-adjusted $p < 0.1$. $N = 10$ Dex, $N = 11$ NoDex.

## Single-cell analysis reveals differing effects of dexamethasone on immune cells from the lung versus blood that are reproducible in external datasets

In order to compare systemic and tissue-specific effects of dexamethasone treatment, we examined single-cell RNA sequencing data from both whole blood and TA from patients treated with or without dexamethasone. We evaluated whole blood (WB) scRNA-seq data from 7 Dex and 3 NoDex, and TA scRNA-seq data from 10 Dex and 7 NoDex patients (Fig. 3a, b). A single data processing pipeline was used to align, harmonize, and cluster data and identify cell types from both compartments (Fig. 3c, d), as well as examine the cell-specific effect of dexamethasone (Fig. 3e, f). Notably, while we include in our gene expression and pathway analysis the cells that are identified as neutrophils, we excluded them from our comparisons of cell type abundance because their proportions were highly discordant with complete blood count results of absolute neutrophil count per white blood cell count (Table S1), likely due to experimental variability in the neutrophil-sparing protocol for scRNA-seq in blood.

Cell-type specific gene expression differences assessed using MAST[17] identified both shared and compartment-specific differential

gene expression associated with dexamethasone (Fig. 3g, h, Fig. S4, Table S3; Supplementary Data File 1). The greatest concordance across compartments appeared in neutrophil differential gene expression (Spearman's correlation R = 0.5; Fig. 3g). Dex subjects exhibited decreases in expression of the S100A family of proinflammatory genes in neutrophils in both lungs and blood. In contrast, gene expression in T cell subsets was highly discordant across compartments (Tregs R = 0.03; CD4 T cells R = 0.05; CD8 T cells R = −0.01; Spearman's correlation). The greatest shared significant difference across anatomical sites in CD4 and CD8 T cells was in the expression of FKBP5 ($log_2$ fold-difference 0.49 and 0.39, and adj. *p*-value 0.023 and 0.058 for CD4 and CD8 T cells, respectively), which is a canonical transcriptomic marker of glucocorticoid receptor activity[18].

In order to assess consistency and reproducibility of our analysis, we also analyzed two external single-cell RNA-seq datasets using this same pipeline: Sinha et al. similarly generated scRNA-seq on whole blood to examine the role of neutrophils in COVID-19 and responsiveness to dexamethasone in an observational cohort of 13 patients (5 Dex/ 8 NoDex)[7]; and Liao et al. acquired bronchoalveolar lavage (BAL) samples from 6 COVID-19 patients[19], a subset of whom were treated

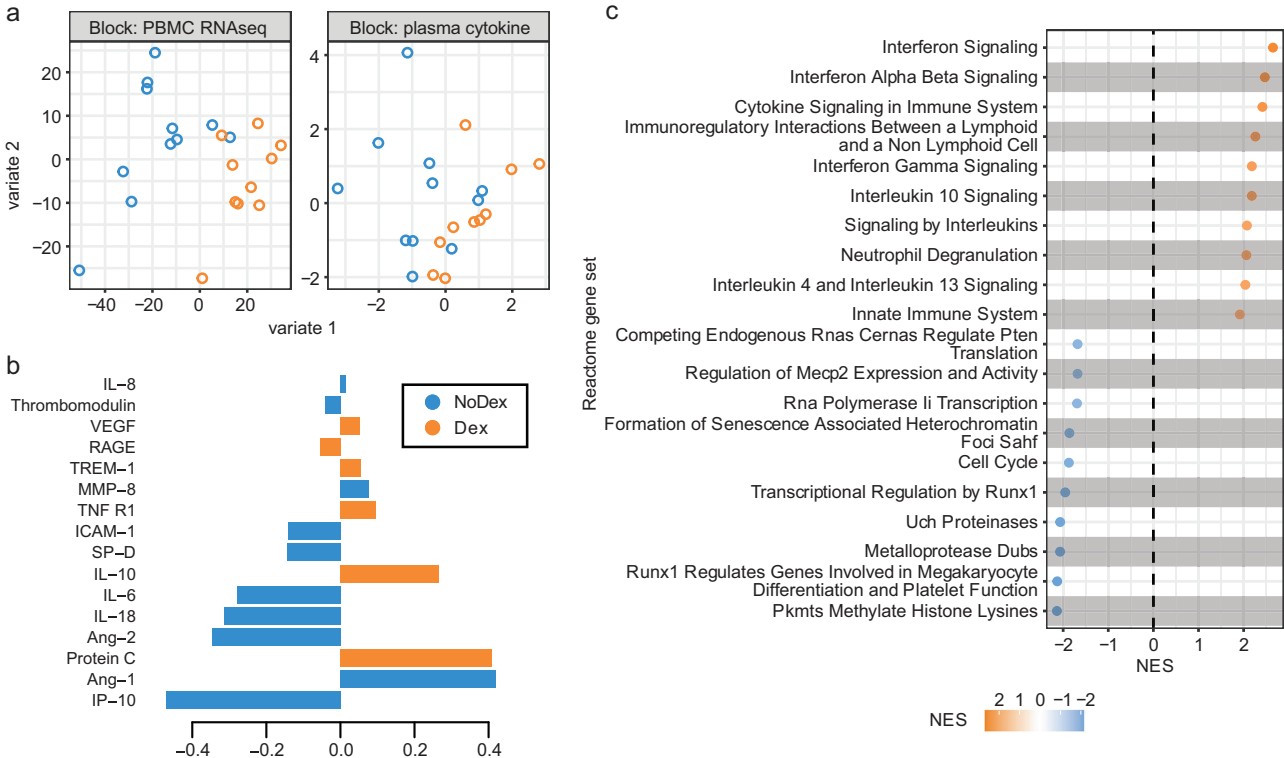

**Fig. 2 | Supervised integrative analysis of blood transcriptomic and plasma cytokine data captures co-varying effects of dexamethasone on immune cell pathways and modulators. a** Integrative analysis of plasma cytokines (17 cytokine variables) and bulk PBMC RNA-seq (500 gene variables) data (paired) from patients comparing Dex and NoDex using DIABLO and highlighting shared contributions from individual data modalities. *N* = 10 Dex, *N* = 11 NoDex; day 0 of hospitalization. First two variates from DIABLO run comparing Dex (orange) vs. NoDex (blue) samples. A parameter value of 0.5 was chosen to model the strength of the relationship between the data and the treatment status. **b** Cytokine contribution (loadings) to DIABLO variate 1. The color indicates the treatment group in which the median value was the highest (orange for Dex and blue for NoDex). **c** GeneNet enrichment scores (NES) from gene set enrichment analysis (one-sided test based on a modified Kolmogorov–Smirnov statistic) of PBMC RNA-seq contribution to DIABLO variate 1 (loadings) using REACTOME gene sets (methods). 20 most significant terms (BH-adjusted *p* < 0.1) represented: top 10 for Dex (orange) and top 10 for NoDex (blue).

with the corticosteroid methylprednisolone (4 methylprednisolone, 2 no-methylprednisolone). Immune cell composition was similar per compartment in external datasets (Fig. S5).

To assess whether the effects of dexamethasone were reproducible across datasets, we tested for enrichment of pathways in the Reactome dataset that were detected across blood datasets (Fig. 4a, Fig. S6) and lung

datasets (Fig. 4b, Fig. S6). In the blood datasets, we observed decreased innate immune signaling and degranulation in neutrophils and decreased immunoregulatory interactions between the lymphoid and non-lymphoid cells in monocytes in Dex patients. Both blood datasets revealed decreased adaptive immune responses and co-stimulation in B cells, as well as decreased levels in cellular responsiveness, and pathways related to infectious disease and influenza responses in both CD4 and CD8 T cells in Dex patients. Interestingly, responses in B cells, CD4 T cells, and monocytes were directionally consistent with a restoration to healthy control levels in these pathways (Fig. 4a, third column), as compared to observations in neutrophils and CD8 T cells.

In contrast, when examining our lung datasets, we observed reproducible but often discordant effects with what was observed in blood, most strikingly an elevation in interferon signaling and response in influenza-related genes in T cell subsets and NK cells in Dex patients that was not observed (interferon) or decreased (influenza) in the blood single-cell datasets (Fig. 4b). Interferon signaling was, as expected, lower in healthy controls than in COVID-19 patients (column 3). Discordant effects also included pathways related to translation and

cellular responses to starvation in CD4 T cells, which appeared higher in lung but lower in blood in Dex patients. Concordant effects across compartments were not detectable.

## Single-cell receptor ligand analysis suggests effects of dexamethasone on tissue injury resolution and a dampening of antigen presentation and T cell responses

Because we identified several differences in cell-specific gene expression, we next sought to understand communication between cells within a compartment to develop a model of the systems biology of dexamethasone in patients with severe COVID-19. We examined ligand-receptor communication using CellChat[20], which extracts signaling patterns among cells from single-cell RNA-seq data. We compared cell-cell signaling between Dex and NoDex subjects in the COMET study patients (blood and TA) and the Sinha et al. study, and compared results against blood scRNA-seq data from healthy controls. In TA, CellChat identified several pathways that were differentially active in Dex and NoDex samples (Fig. 5a). Dexamethasone was associated with a marked decrease in MHC-II signaling (Fig. 5b), suggesting a potential decrease in antigen presentation to CD4 cells in the lung. In addition, CellChat identified a significant decrease in SELPLG activity in TA (Fig. 5c), suggesting dexamethasone might play a role in decreasing lung injury through these mechanisms, given prior studies associating SELPLG with murine lung injury and higher risk for non-COVID-19 ARDS in humans. Similar effects were also observed in blood, but the effect was much smaller in magnitude than in TA samples and statistically insignificant.

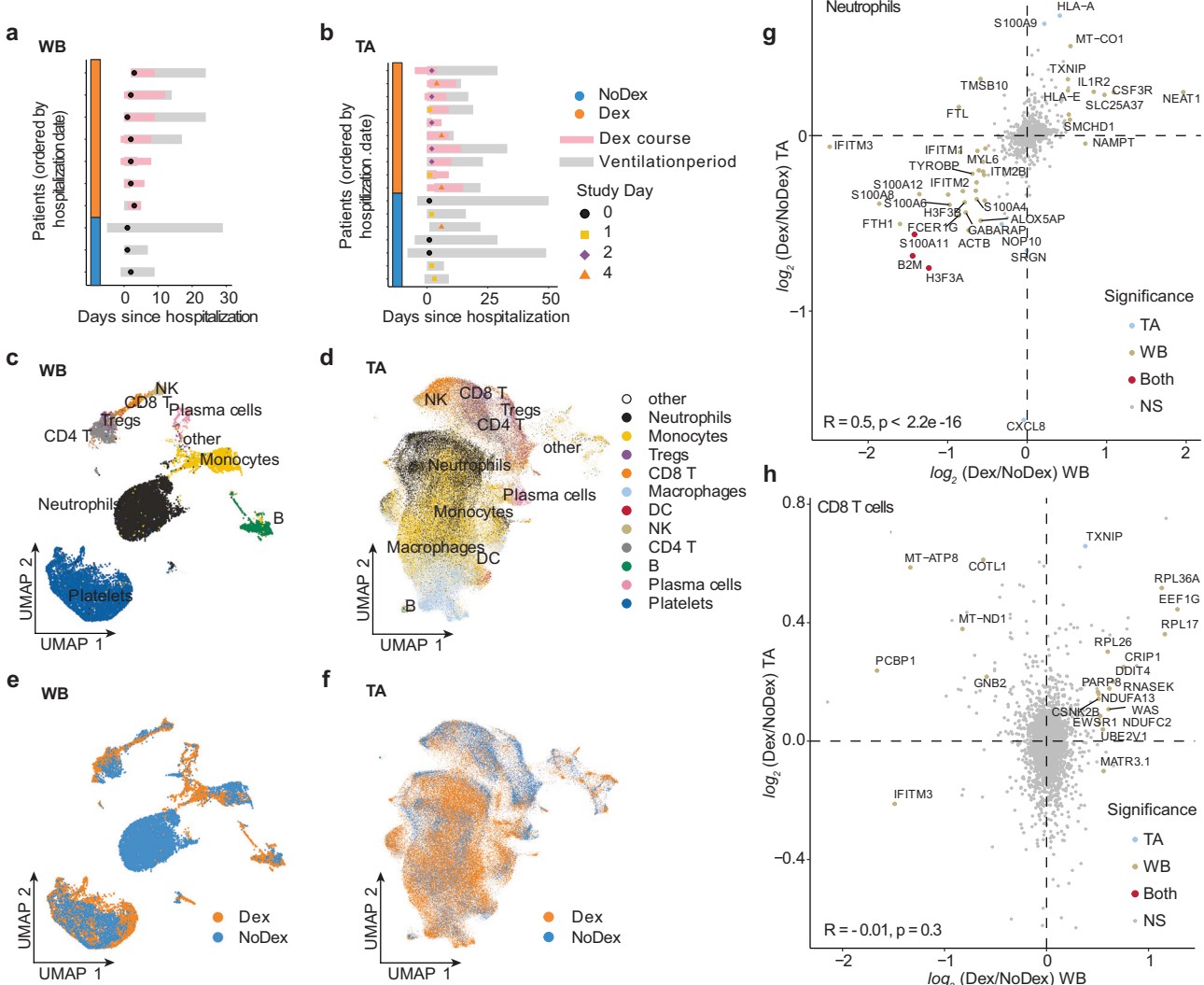

**Fig. 3 | Single-cell analysis of lung and peripheral blood samples from patients treated with or without dexamethasone.** Plot per patient showing the collection of whole blood (WB) (**a** N = 7 Dex, 3 NoDex) or tracheal aspirate (TA) samples (**b** N = 10 Dex, 7 NoDex) overlaid on hospitalization (gray bars) and dexamethasone treatment (pink bars). *X*-axis shows days of hospitalization (day 0 = admission to UCSF hospital). Dots show the day when sample was collected, colored by Study Day (methods). UMAP plots of single-cell RNA-seq data from blood (**c**) or TA (**d**) samples, clustered and annotated by major immune cell types. UMAP plots of single-cell RNA-seq data from blood (**e**) or TA (**f**) samples, colored by Dex (blue) or NoDex (pink) samples. *log₂* fold difference of gene expression of Dex and NoDex in TA (*y*-axis) v. blood (*x*-axis) plotted for Neutrophils (**g**) and Tregs (**h**). Significant genes in TA only (blue), blood only (brown), both compartments (red) are shown (BH-adj. $p < 0.1$ & |*log₂* fold-difference | > 0.5). Spearman's correlation *R* value shown between the two compartments.

Dexamethasone was associated with additional differences in whole blood that were consistent with findings in the Sinha et al. dataset. A clustered heatmap of detected interactions grouped together the two NoDex COVID-19 datasets, whereas the two Dex COVID-19 datasets grouped with each other and with the healthy control dataset, suggesting dexamethasone may be contributing to a restoration toward a healthy phenotype (Fig. 5d). The collagen and annexin pathways were more active in NoDex subjects, and activity of these pathways in Dex subjects was comparable to healthy controls (Fig. 5e, Fig. S7). Interestingly, collagen deposition can occur in the context of viral infection, likely as a response to injury and inflammation, and the restoration to healthy control levels may further indicate reduction of that response. In addition, elevation of CD99, ICAM, and ITGB2 were observed in NoDex patients compared to both Dex patients and healthy controls (Fig. 5e, Fig. S7). This finding may indicate an effect of dexamethasone on dampening T cell responses since these signaling molecules are involved in leukocyte recruitment, formation of the immunological synapse between T cells and antigen presenting cells, and T cell function and activation[21].

## Discussion

Despite their widespread use in clinical medicine and demonstrated benefit in patients with severe COVID-19 infections, the biological effects of corticosteroids on pulmonary and systemic biology in critically ill patients are incompletely characterized. We performed a multi-omic analysis of the effects of dexamethasone in a cohort of patients with severe COVID-19. We identified cell- and compartment-specific effects of dexamethasone that highlight the pleiotropic effects of steroids in critical illness. Limited data are available about the compartmentalized biological effects of steroids in patients with ARDS, pneumonia, or sepsis due to causes other than COVID-19, and the role of corticosteroids in treating these conditions in patients remains uncertain[22–24]. Our analysis identifies dysregulated pathways potentially modified by dexamethasone therapy that could have potential therapeutic relevance in other causes of critical illness[25].

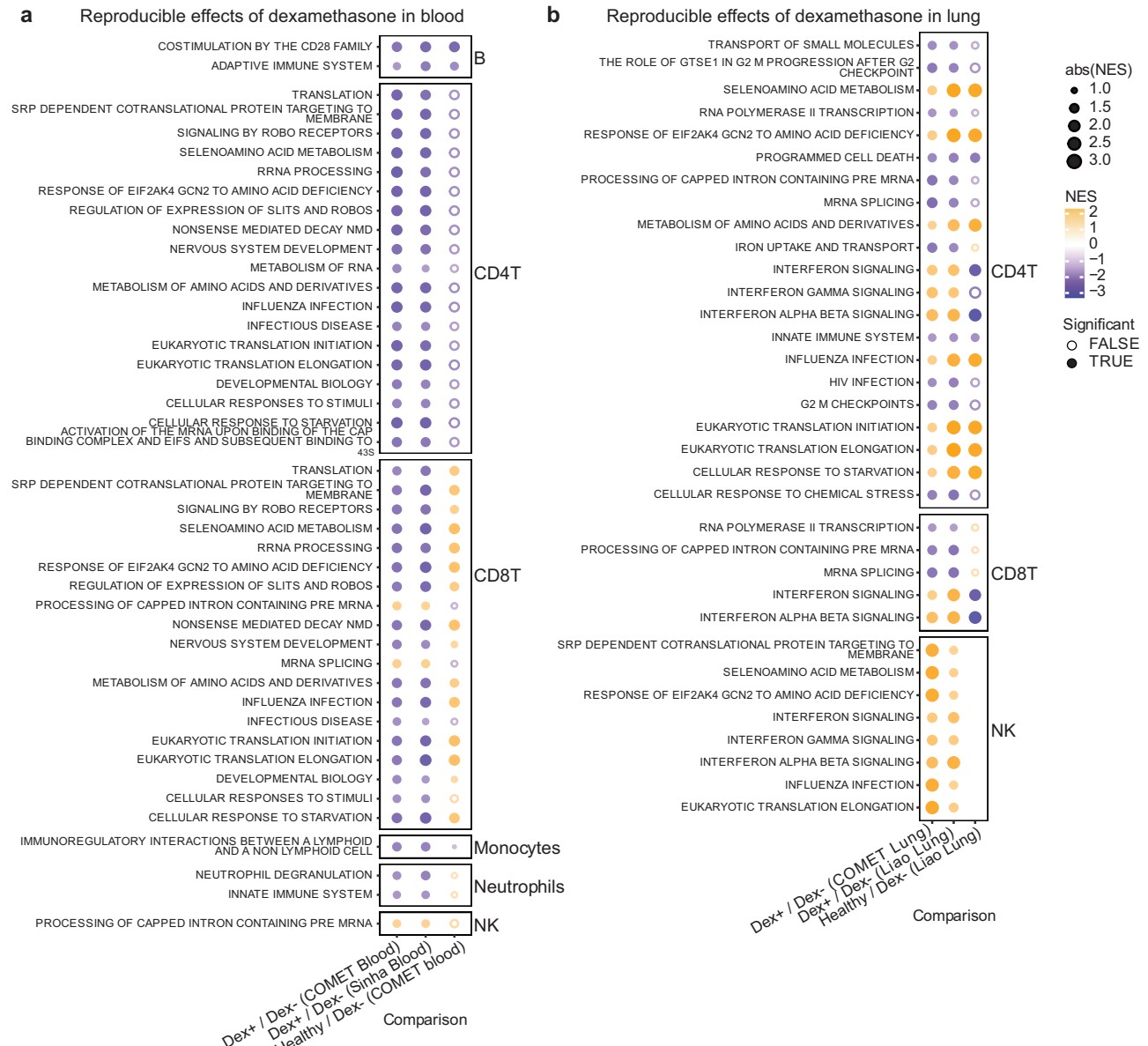

**Fig. 4 | Dexamethasone has discordant effects on cell type specific gene expression in lung and peripheral blood that are reproducible in external datasets.** Net enrichment scores from gene set enrichment analysis in blood (**a**) and lung (**b**), faceted by cell type. Orange circles have a positive net enrichment score (NES), indicating the pathway is more highly expressed in dexamethasone-treated COVID-19 patients (Dex) or healthy controls relative to NoDex subjects.

Solid circles identify pathways where GSEA BH-adjusted *p* < 0.1, empty circles identify pathways with GSEA BH-adjusted *p* ≥ 0.1, and blank spaces indicate no GSEA NES score was calculated for that pathway. Significance was determined using a one-sided test based on a modified Kolmogorov–Smirnov statistic. Datasets represented are from COMET (whole blood, TA), Sinha et al. (blood) and Liao et al. (BAL). Ns reported in Fig. S8 and Supplementary Data File 2.

Integrative analysis of cytokine and blood transcriptomics identified decreased plasma concentrations of IP-10 in Dex patients. IP-10 is an interferon-stimulated molecule that promotes T-cell adhesion to endothelial cells[26], and has been associated with disease severity and mortality in COVID-19 patients[27]. Consistent with this result, interferon-gamma concentrations were also lower in patients treated with dexamethasone. In contrast to IP-10 and IFN-gamma protein levels, interferon-stimulated genes were markedly upregulated in dexamethasone-treated patients in our integrative analysis. The discordance between interferon levels from protein biomarker data and the enrichment of interferon-related genes may reflect steroid-resistant ISG pathways remaining active in these patients, which may explain the efficacy of JAK/STAT inhibition in patients treated with steroids[28]. We also found higher levels of Ang-1, and lower concentrations of its antagonist, Ang-2, were associated with

dexamethasone treatment. An increased ratio of Ang-2 to Ang-1 reflects endothelial injury[29], and is associated with mortality in patients with ARDS due to COVID-19 and other causes[30]. Together, the results of our integrative analysis demonstrate treatment with dexamethasone is associated with decreased activation of several pathways associated with COVID-19 severity.

Inference and analysis of cell communication identified potential cellular signaling networks that may explain changes in COVID-19 biology associated with dexamethasone treatment, including decreased antigen presentation, leukocyte recruitment and activation, and signatures of tissue injury. In TA, dexamethasone treatment was associated with decreased activity of MHC-II and SELPLG, a glycoprotein involved in leukocyte trafficking in inflammation. Notably, *SELPLG* was identified as a locus associated with increased risk of ARDS in GWAS studies, pulmonary *SELPLG* expression is increased in murine

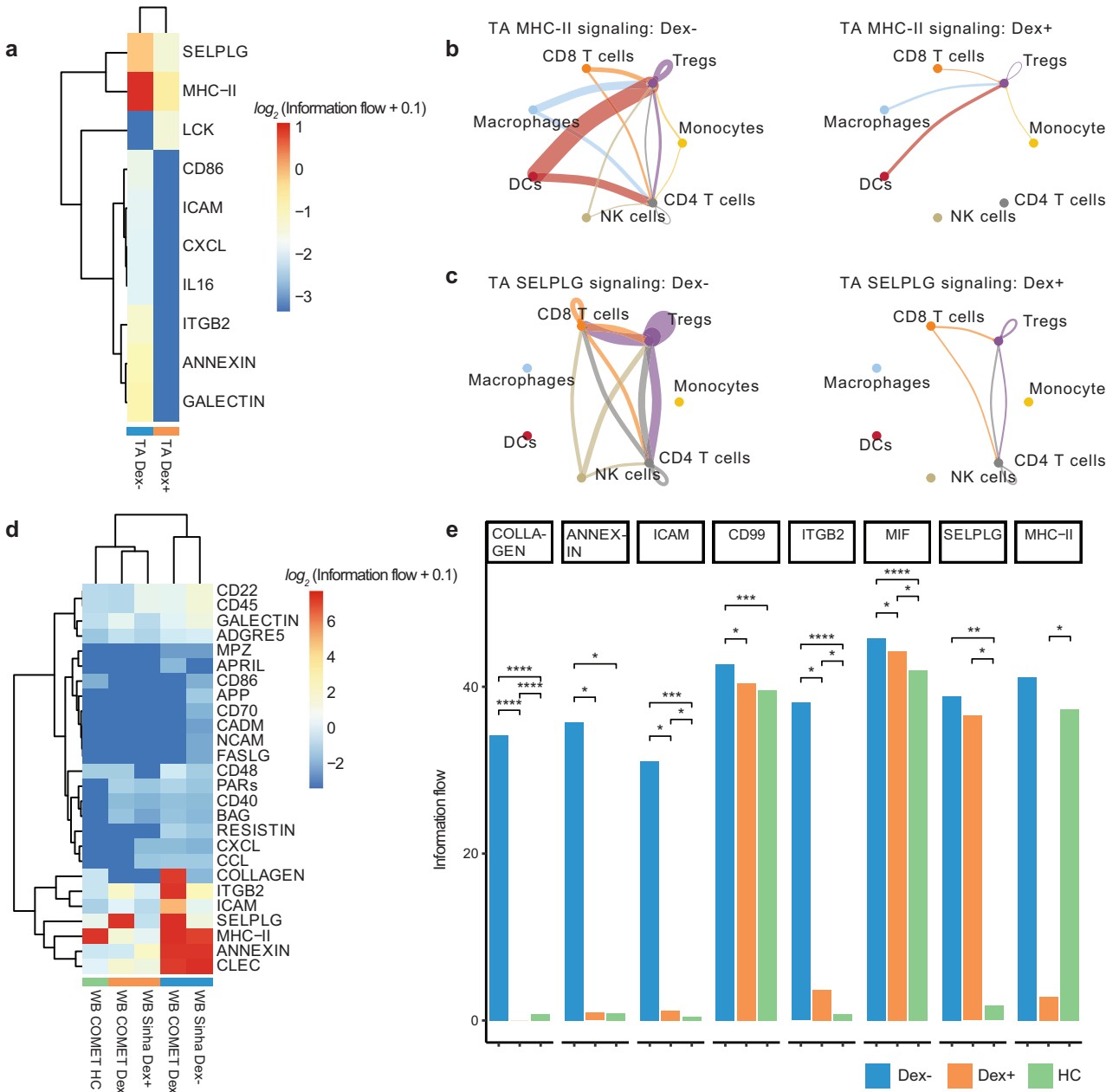

**Fig. 5 | Receptor ligand inference from single-cell sequencing data reveals decrease in inflammation, antigen presentation, and T cell recruitment in blood and lung in response to dexamethasone. a** Clustered heatmap of CellChat results of TA samples from Dex ($N = 10$) and NoDex ($N = 7$) patients with significant receptor-ligand pairs shown (based on one-sided Wilcoxon signed rank test (BH-adj. $p < 0.1$ and $|log_2$ fold-difference$| > 1$). Cell type interaction networks for MHC-II (**b**) and SELPLG interactions (**c**) shown comparing NoDex (left, $N = 7$) and Dex (right, $N = 10$) patients of TA samples. Line thickness represents predicted strength of the interaction. **d** Clustered heatmap of CellChat results of blood samples from Dex (COMET), Dex (Sinha et al.), NoDex (COMET), NoDex (Sinha et al.), and healthy controls (COMET) with receptor-ligand pairs that are significant between at least one pair of patient groups are shown (based on one-sided Wilcoxon signed rank test (BH-adj. $p < 0.1$ and $| log_2$ fold-difference $| > 1$). **e** Comparisons for the COMET dataset shown between Dex, NoDex, and healthy controls for a subset of significantly detected receptor-ligand interactions (*adj. $p < 0.1$, **adj. $p < 0.001$, ***adj. $p < 0.0001$, ****adj. $p < 0.00001$; BH-adjusted). Ns reported in Fig. S8 and Supplementary Data File 2.

lung injury models, and anti-SELPLG antibodies decrease LPS-induced lung injury[31]. In both the respiratory tract and whole blood, dexamethasone was associated with decreased MHC-II activity. Dexamethasone inhibits expression of MHC-II in dendritic cells in experimental models[32], which may further suppress immune responses by decreasing antigen presentation to T cells. Decreased co-stimulation as evidenced by reduced expression of TNFRSF4 (OX40, Fig. 1d), gene sets (B cell co-stimulation pathway, Fig. 4), and receptor-ligand co-expression (CD86 signaling, Fig. 5) further supports this

model of resultant decreased T cell activation via impaired antigen presentation.

Network analysis of whole blood scRNA-seq data revealed decreased activity of annexin, integrin beta 2, and ICAM pathways, which mediate leukocyte adhesion and extravasation. These decreases were also observed in TA. Annexins play a key role in resolving inflammation and are established glucocorticoid targets[33]. Beta2 integrins are adhesion molecules that regulate neutrophil function, and leukocyte adhesion and trafficking. Our results are consistent

with prior observations that steroids decrease the expression of integrin beta 2 (CD18) in activated neutrophils[34]. Intercellular adhesion molecules enable leukocyte recruitment to injured lung and, in patients with non-COVID-19 ARDS, increased concentrations of sICAM-1 are associated with a higher mortality, hyperinflammatory ARDS phenotype[35,36] and dexamethasone also inhibits LPS-stimulated ICAM-1 signaling[37]. ICAM-1 has additionally been reported to be higher in non-survivors than survivors of COVID-19 related ARDS[11]. In whole blood, we also observed decreased activity of collagen pathways with dexamethasone treatment, which may reflect a mitigation of damage from viral injury[38]. The results of the network analysis identify several dysregulated cell-signaling pathways that may be modified by dexamethasone treatment and mediate the therapeutic effects of steroids in each the lungs and blood.

This study significantly adds to prior studies of the effects of steroids in patients with COVID-19 by identifying reproducible cell-specific and compartment-specific effects of dexamethasone. Prior observational studies have identified changes in neutrophilic inflammation and gene expression associated with corticosteroids in patients with COVID-19. Steroids were associated with decreased BAL neutrophils in a case series of 12 patients with COVID-19 ARDS who required ECMO[39]. In patients with non-resolving ARDS, steroid treatment was associated with decreased BAL concentrations of the neutrophil chemoattractants CXCL1 and CCL20[40]. Two observational studies have described the effects of dexamethasone on gene expression in patients with COVID-19 ARDS. Sinha et al. compared peripheral scRNA-seq data from six dexamethasone-treated patients to eight controls, and found that dexamethasone was associated with decreased annexin signaling, increased circulating immature neutrophils, and suppression of interferon-stimulated neutrophils[7]. The second compared bulk RNA sequencing in BAL samples from eight patients treated with dexamethasone to four who did not receive dexamethasone, and identified genes that were differentially expressed between the groups related to B cell activation, leukocyte trafficking, and antigen presentation[8]. Our results build on these prior studies, and support a model of dexamethasone mitigating excess inflammatory damage in severe COVID-19 by reducing the immune response to viral infection via decreased antigen presentation and T cell recruitment and activation.

Our results suggest dexamethasone has distinct effects on pulmonary and systemic inflammation and repair in patients with COVID-19, consistent with prior findings from lung injury models. Michel et al. challenged healthy volunteers with inhaled LPS and observed an increase in sputum and peripheral blood inflammatory biomarkers. Prednisolone 10 mg had no effect on airway inflammation but markedly decreased plasma CRP concentrations[41]. Bartko et al. bronchoscopically instilled LPS into lung segments of healthy volunteers and saline into a contralateral segment. Pretreatment with 40 mg of dexamethasone 13 h and 1 h before LPS challenge markedly decreased systemic inflammation biomarker levels, BAL neutrophilia, and BAL protein concentrations, but only minimally decreased BAL IL-6 concentrations and had no effect on BAL TNF or IL-8 concentrations[5]. We observed several cell- and compartment-specific differences in gene expression associated with dexamethasone treatment, emphasizing the importance of studying respiratory illness biology not only systemically, but also at the site of injury.

This integrated analysis should inform the design of future clinical, translational, and basic studies of COVID-19 biology. We observed decreased signatures of T cell activation in patients who were treated with steroids. This likely suppresses viral clearance, which may explain the increased viral antigen in COMET patients treated with steroids and higher viral loads in experimental models[42]. This suppression of antiviral responses may also explain the trends toward increased mortality in patients who received steroids before they required supplemental oxygen in the RECOVERY dexamethasone trial[2] and is an

important consideration the design of future studies of steroids in patients with respiratory infections. While T cell activation was relatively suppressed, immune signaling pathways remained dysregulated in COVID-19 patients who received steroids compared to healthy controls, which may explain why patients who are treated with dexamethasone still benefit from additional immunomodulatory therapies. In the RECOVERY platform trial, tocilizumab, an IL6-receptor antagonist, decreased mortality by 15% in a cohort where 82% of patients received steroids, and baricitinib, a JAK inhibitor, decreased mortality by 20% in a cohort where 95% received steroids[28,43]. Further study is required to understand whether therapies for pathways that remain dysregulated after steroids (e.g., with selectin P ligand pathway inhibitors) can further improve outcomes in COVID-19. Steroids also have numerous adverse effects, and more narrowly targeted treatments that treat pathways modified by steroids may provide the same benefit with fewer adverse effects.

This study has several strengths. We selected subjects from a deeply phenotyped observational cohort and integrated multiple assays to identify compartment- and cell-specific differences in the responses to dexamethasone. We build on prior studies by examining both the systemic and pulmonary biology of COVID-19 together, which provides more complete insight into the pathophysiology of critical respiratory illness. We used mixed effects modeling to compare single cell RNA expression, which addresses the pseudo-replication bias present in prior clinical single cell studies and produces more conservative and reproducible estimates of differential gene expression. Our findings extend our understanding of corticosteroids in critical respiratory illnesses, at the gene, protein and cellular levels. Future studies using similar methods can assess whether these observations are generalizable to patients with other critical illness syndromes, such as sepsis or ARDS.

This study also has some limitations. COMET is an observational study, so treatment with dexamethasone was not randomly assigned, and we cannot rule out confounding by other unobserved variables that also changed during the study period. However, we carefully selected patients for inclusion in both the Dex and NoDex cohorts to minimize the effects of practice variation (Methods). As discussed above, we also observed higher plasma N-antigen concentrations in COMET patients who received dexamethasone. While Dexamethasone notably impairs viral clearance in experimental models of SARS-CoV2 pneumonia[42], we cannot confirm steroids are responsible for this effect in our cross-sectional dataset. Reassuringly, many of our observations are reproducible in external cohorts and are consistent with experimentally confirmed effects of dexamethasone. Secondly, it is challenging to temporally align specimens from critically ill patients, who have dynamic and rapidly changing biology. This variance can introduce additional within-group biological heterogeneity and bias comparisons toward the null; despite this challenge, we were able to identify robust and reproducible signals using multiple modalities, suggesting the date of intubation was a suitable reference timepoint for sample collection. Because this was an observational, cross-sectional study, we cannot determine if differences in cell- and compartment-specific gene expression represent proliferation of cell lines, changes in cell polarization, and/or translocation of cells between the pulmonary and systemic compartments.

In summary, we identified cell- and tissue-specific differences in the effects of dexamethasone in critically ill patients with COVID-19. This includes a reproducible increase in interferon gene signatures in dexamethasone-treated patients, potential evidence of steroid-resistant interferon gene signaling in response to a putative higher viral load in dexamethasone-treated patients. We see evidence of decreased antigen presentation (MHCII pathway signaling) and reduced co-stimulation genes and pathways, paired with signatures of decreased T cell recruitment and activation, which may contribute to dexamethasone's effect of reduced immune-related damage. Our

results provide new insights into potential therapeutic targets in COVID-19 and highlight the importance of studying compartmentalized immune responses in critically ill patients.

## Methods

### Study

We conducted a case-control study of mechanically ventilated COVID-19 ARDS patients with (Dex) or without (NoDex) administered dexamethasone. The patients used in this study were a subset of the participants enrolled in the COMET study (COVID-19 Multi-immunophenotyping projects for Effective Therapies https://www.comet-study.org/), which had a partial overlap with the IMPACC (IMmunoPhenotyping Assessment in a COVID-19 Cohort)[9]. Some of the data derived from these patient samples have been presented in prior publications from the IMPACC and COMET consortia[10,44,45] and some is presented here for the first time, which is delineated per modality in the following sections and further specified in Supplementary Data File 2 and Fig. S8. These patients were enrolled either at the University of California, San Francisco Medical Center (UCSFMC) and Zuckerberg San Francisco General Hospital (ZSFG). The COMET study was approved by the UCSF Institutional Review Board (IRB #: 20-30497). We included patients who were enrolled between April 2020 and Mar 2021. The NoDex group ($n = 16$) included patients enrolled before July 2020, when the dexamethasone became the standard of care for COVID-19. The Dex group ($n = 27$) included patients enrolled after July 2020. The patients were enrolled in a study within the first 72 h of hospitalization. The blood samples were collected on the day of enrollment ("Study Day 0") and tracheal aspirates were collected within four days of enrollment. We selected only a single timepoint per patient in each assay for this study.

### Subjects

**Consent.** Informed written consent was obtained from subjects or their surrogates. If a patient was unable to consent due to critical illness and a surrogate was unavailable, the UCSF IRB granted a waiver of initial consent. For all subjects included in this study, informed consent was either obtained at enrollment, or follow-up consent was then obtained from the patient if they regained the ability to consent, or from a surrogate if they did not.

As the COMET database is regularly updated, we chose to freeze our list of included patients based on a snapshot of the database as of May 9, 2022. To be selected, patients had to meet all following criteria: confirmed COVID-19 infection; ICU admission record or WHO COVID-19 severity score of 6 or more at any point during hospital stay; not on an immunosuppressive therapy; for dexamethasone-treated patients, not be on a different steroid with an overlapping range, or prior admission; complete and unambiguous treatment record available; and intubated (Table S1, Fig. S1).

### Data acquisition

**Luminex assay for plasma cytokines.** The soluble plasma cytokines were quantified using the Luminex multiplex platform (Luminex, Austin TX)[10]. Briefly, the analytes were quantified using the Luminex multiplex platform with custom-developed reagents (R&D Systems, Minneapolis, MN), as described in detail[46] or single-plex ELISA (R&D Systems, Minneapolis, MN). The quantified analytes were read on MAGPIX® instrument and the raw data was analyzed using the xPONENT® software. Analytes quantified using single-plex ELISA were read using optical density. Values outside the lower limit of detection were imputed using 1/3 of the lower limit of the standard curve for analytes quantified by Luminex and 1/2 of the lower limit of the standard curve for analytes quantified by ELISA.

**Bulk RNA sequencing of PBMCs.** The bulk RNA sequencing library preparation for PBMC was performed using SMART-Seq Low Input

protocol for all samples in COMET and IMPACC studies as described here[44]. Briefly, RNA was extracted from $2.5 \times 10^5$ PBMCs using the Quick-RNA MagBead Kit (Zymo) with DNase digestion. RNA quality was assessed using a Fragment Analyzer (Agilent) and 10 ng RNA was used to synthesize full length cDNA using the SMART-Seq v4 Ultra Low Input RNA Kit (Takara Bio). The cDNA was purified using bead cleanup, followed by library preparation using Nextera XT kit (Illumina). Libraries were validated on a Fragment Analyzer (Agilent), pooled at equimolar concentrations, and sequenced on an Illumina NovaSeq6000 (Emory) at 100 bp paired-end read length targeting ~25 million reads per sample. Data for ten out of the 21 samples used here were obtained from an IMPACC publication[44]. The details on individual samples are included in Supplementary Data File 2.

**Single-cell RNA sequencing of TA and WB.** The single cell RNA sequencing of TA and WB samples was performed for all samples in the COMET study[10,45]. Briefly, the TA samples were transported to a BSL-3 laboratory, 3 mL of TA was dissociated using 50 µg/mL collagenase type 4 (Worthington), and 0.56 ku/mL of Dnase I (Worthington). The single-cells were collected by centrifugation and counted, and the CD45-positive cells were enriched using MojoSort Human CD45 beads (Biolgenend) and counted again before library preparation. The scRNA-seq of whole blood was performed to preserve granulocytes. Briefly, the peripheral blood was collected into EDTA tubes (BD, 366643). 500 µl of peripheral blood was treated with RBC lysis buffer (Roche, 11-814-389-001) according to the manufacturer's instructions and the single cells were collected and counted. For both TA and WB samples, the Chromium Controller was loaded with 15,000 cells per sample following the manufacturer's instructions (10X Genomics). Some samples were pooled together (at 15,000 cells per sample) before GEM partitioning. A Chromium Single Cell 5′ Reagent Kit v2 (10X Genomics) was used for reverse transcription, cDNA amplification and library construction of the gene expression libraries (following the detailed protocol provided by 10X Genomics). Libraries were sequenced on an Illumina NovaSeq6000. Data for three out of the 10 WB samples from COVID-19 patients and all WB samples from healthy controls were obtained from a previous COMET publication[10]. Data for three out of 17 TA samples from COVID-19 patients were obtained from another COMET publication[45]. The details on individual samples are included in Supplementary Data File 2.

### Cytokine analysis

Cytokine data was represented using principal component analysis. For this analysis only, variables with more than 10% missing values across the dataset were excluded. Patients with one or more remaining missing values were filtered out. Values were then $log_2$-transformed and scaled. A PERMANOVA test was performed using Euclidean distances to estimate separation of the treatment groups. To compare circulating cytokine levels, Wilcoxon rank-sum tests on cytokine concentrations, including those with more than 10% missing values, were employed. Significant differences were selected using a 0.1 threshold on BH-adjusted $p$-values.

### Bulk RNA sequencing analysis

Gene counts were generated using the nf-core *rnaseq* pipeline v3.3 (https://nf-co.re/rnaseq) and Salmon-generated counts were used for the analyses.

For the analysis of bulk gene expression data, the R package DESeq2 (v1.28.1) was used. Age and sex assigned at birth were included as covariates in the model. The *log* fold-change values were shrunk using the *apeglm* algorithm. A 0.1 threshold on BH-adjusted $p$-values was used to identify differentially expressed genes. Gene set enrichment analysis was performed on the full list of genes sorted by shrunken *log* fold-change values with the *fgsea* package (v.14.0) and the REACTOME gene set database. Significantly

disrupted pathways were identified using a 0.1 threshold on BH-adjusted *p*-values.

## Integrative analysis

DIABLO (v6.14.11), a supervised multi-omics data integration tool, was selected to analyze coordinated changes across cytokine and bulk PBMC data, and to identify variables driving the differences between NoDex and Dex patients. DIABLO extends sparse generalized canonical correlation analysis (sGCCA) to a supervised framework. sGCCA uses singular value decomposition and selects correlated variables across several omics datasets, such that the covariance between linear combinations of variables (latent component) is maximized.

Only intubated patients with both cytokine and bulk PBMC data measurements were selected for the integrative analysis. Scaled *log₂* transformed cytokine values and scaled variance stabilization transformed counts for the 500 most variable genes were used as input. DIABLO's parameter *design* (range 0–1) indicates the extent to which covariance between data modalities should be maximized vs. covariance between individual data modalities and treatment status. We chose a value of 0.5 to balance the contribution of those two covariances for our analysis.

## Single-cell RNA sequencing analysis

**Data processing.** The BCL files from sequencer were demultiplexed into individual libraries using mkfastqs command in Cellranger 3.0.1 suite of tools (https://support.10xgenomics.com). The feature-barcode matrices were obtained for each library by aligning the WB raw FASTQ files to GRCh38 reference genome (annotated with Ensembl v85) and TA raw FASTQ files to GRCh38 + SARS-CoV-2 reference genome using Cellranger count. The raw feature-barcode matrices were loaded into Seurat 4.0.3, and cell barcodes with minimum of 100 features were retained in order to remove the droplets lacking cells. The features that were detected in less than 3 barcodes were removed, and the expression data was log-normalized using NormalizeData() function within Seurat. Our dataset contained three samples that were multiplexed for 10X library preparation and the rest were processed individually. For the samples that were processed individually, the heterotypic doublets were detected using DoubletFinder[47] by matching each cell with artificially synthesized doublets. We used 35 PCs, pN=0.25 and sct=TRUE in DoubletFinder. An optimal pK value (PC neighborhood size used to compute pANN) was determined for each sample separately using find.pK function as suggested by the authors. We approximated the doublet rate as 7% based on 10X's recommendation for the expected doublets when 15,000 cells were loaded on the 10X handler (https://kb.10xgenomics.com/hc/en-us/articles/360001378811). DoubletFinder requires cell annotations to determine the rate of heterotypic doublets. We clustered the cell barcodes using Louvain clustering and the cluster labels were used as cell annotations. We removed the heterotypic doublets and subjected the remaining barcodes for further quality control.

Our dataset contained three samples that were multiplexed, for which the filtered count data for singlets were obtained from GSE163668[10]. The authors used Demuxlet[48] to demultiplex the samples and to identify inter-sample doublets, and DoubletFinder to identify heterotypic doublets. Single cells with greater than 50,000 unique RNA molecules, fewer than 150 or greater than 8000 features, greater than 15% mitochondrial content or greater than 60% ribosomal content were removed. The cell cycle state of each cell was assessed using a published set of genes associated with various stages of human mitosis[49].

The WB data from healthy controls was obtained from GSE163668[10], the external validation WB data from COVID-19 patients from GSE157789[7] and the external validation bronchoalveolar lavage (BAL) fluid data from GSE145926[19]. The same data processing strategy was used for these datasets as for our datasets described above.

**Data integration and UMAP generation.** There was a substantial heterogeneity between samples within treatment groups, most likely due to technical variations introduced during the library preparation that spanned over months. Even if this heterogeneity is due to biological differences, this heterogeneity could cause substantial issues in mapping same cell types across samples. To account for this, we integrated the samples using Seurat's CCA integration approach (FindIntegrationAnchors and IntegrateData functions)[50], while treating each sample as its own batch. The integrated data was scaled while regressing out feature counts, RNA counts, mitochondrial percentage, ribosomal percentage and cell states. After reducing the data to lower dimensions (PCs), 30 PCs were used for UMAP generation. The CCA integrated data was used only for generating UMAPs. All follow-up analyses were performed using the non-integrated data. Each tissue was processed separately.

**Single-cell annotation.** Automated cell annotation was performed using SingleR[51]. We mapped the *log*-normalized expression data against a reference expression dataset from ENCODE Blueprint[52]. The fine labels of Blueprint dataset were used for mapping. Many cell types contained too few cells, which were cleaned up in two ways: the cell types with less than 101 cells across all samples from a tissue were labeled "other" and fine labels were manually combined into broad cell types for the follow up analyses.

**Differential frequency analysis.** The cell frequencies were normalized to the total cell counts per sample, multiplied by 100, and compared between Dex and NoDex samples using Wilcoxon rank-sum test. The *log₂* fold-difference was calculated by calculating the *log*-ratio of mean normalized frequencies of Dex and NoDex samples after adding 0.01 to all values to avoid log-of-zero errors. The Neutrophils were removed before frequency normalization. No adjustment of *p*-values was performed here.

**Differential gene expression.** To study the cell-type-specific effects of dexamethasone in whole blood and TA samples, we compared gene expression between Dex and NoDex samples within each tissue for every cell-type separately. The differential expression analysis was performed using Model-based Analysis of Single-cell Transcriptomics (MAST)[17]. MAST models single cell expression data using a hurdle model with the expression data represented as $log_2$(transcripts per million + 1). The hurdle model is a two-part generalized linear model that simultaneously models the rate of signal over background and continuous expression level, conditional on the gene being expressed. Specifically, the expression level is modeled as a truncated Gaussian distribution. This approach allows for controlling for both the non-detectable signal (zero inflation; which is prevalent in scRNA-seq datasets) and the non-zero expression. The MAST analysis was performed in the following manner. The cell types with at least 50 cells in both conditions were retained in the Seurat objects. For each cell type, the Seurat object was subsetted to keep single-cell expression data for that cell type, the subsetted object was converted to SingleCellExperiment object, and the RNA raw counts were normalized for the library size (i.e., divided each count by total number of UMIs per cell and multiply by the mean of the number of UMIs per cell across all cells) and $log_2$ transformed with pseudocount of 1. To remove the highly sparse data, only genes with non-zero counts in at least 5% cells in at least one condition were retained. Finally, the zlm function was used to identify the differentially expressed genes between Dex and NoDex samples. We also accounted for the number of detected genes per cell by adding it as a covariate in the model. Because transcriptomes of cells from the same sample are not independent observations, we included patient IDs as a random effect to account for the hierarchical nature of the data and prevent pseudoreplication bias. Additionally, we used the following parameters in zlm function: method = 'glmer', ebayes = F, strictConvergence =

FALSE, fitArgsD = list(nAGQ = 0). Finally, the *P* values were corrected for multiple testing using the BH procedure.

**Gene set enrichment analysis.** To identify the pathways affected by the dexamethasone treatment, we performed gene set enrichment analysis (GSEA) on the full list of genes[53]. GSEA tests whether a pre-selected set of genes is distributed randomly in a ranked gene list (the null hypothesis) or is concentrated on one end of the list (the alternative hypothesis). The test statistic and the resulting *p*-value recapitulate the likelihood of observing a given enrichment score considering a distribution of simulated enrichment scores obtained by random permutation of gene ranks. We used fgsea in R which is a collection of tools allowing fast gene set enrichment analysis. We ranked the genes by shrunken $log_2$ fold-changes between pairs of Dex, NoDex and healthy samples and used the fgseaMultilevel function (nPermSimple = 10000 and minSize = 25) to perform GSEA analysis against REACTOME pathways. Significantly different pathways were identified using a 0.1 threshold on BH-adjusted *p*-values.

**CellChat analysis.** CellChat[20] was used to identify ligand-receptor pairs that display differential interaction strength between cells from Dex, NoDex and healthy groups. First, differentially expressed signaling genes were identified across groups using the Wilcoxon rank-sum test. Next, probabilities for each interaction were estimated using an equation derived from the law of mass action. Finally, the significance of those probabilities was estimated using permutations. Each permutation was generated by randomly shuffling phenotype labels (Dex and NoDex) and computing probabilities that were then used to estimate the likelihood of the observed value. The Seurat objects were subsetted to include the cell types that had more than 100 cells in all conditions within that tissue. Specifically, for TA data, the cell types with more than 100 cells in both Dex and NoDex were retained, and for blood data, the cell types with more than 100 cells in all groups (Dex (COMET), NoDex (COMET), healthy (COMET), Dex (Sinha et al.) and NoDex (Sinha et al.)) were retained. The CellChat objects were first created for each group (condition) of cells separately using create-CellChat() function, with Seurat's normalized RNA data as input data. The over expressed genes and interactions were identified based on the CellChat database of human ligand-receptor pairs, and the expressed data were projected on the protein-protein interaction network. Finally, the communication probabilities were calculated, the communications based on less than 10 cells were discarded, aggregated network were calculated by summarizing the communication probability, and saved as individual RDS files for each condition. Pairs of conditions, for example TA Dex and TA NoDex, were compared using rankNet to rank signaling networks based on the information flow. The rankNet uses Wilcoxon signed-rank test to identify pathways that have significantly different communication probabilities between a pair of conditions across cell types. We used this information flow to find ligand-receptor pairs that exhibit significant difference in predicted interaction strength between the conditions.

**Statistics**

Statistical comparisons were performed using one-sample Wilcoxon rank-sum test or Chi-squared test, for continuous and categorical variables, respectively, unless mentioned otherwise. All *p*-values, including the ones derived from independent tools were corrected using the same procedure i.e., using the Benjamini–Hochberg (BH) method, which controls for the false-discovery rate, unless stated otherwise. We used the p.adjust method in R using the "BH" option, A threshold of 0.1 was chosen to determine significance. All correlation *R* values are calculated using Spearman's correlation.

Additional analyses with associated statistical tests were performed using the MAST[17], fgsea[53], CellChat[20], and DIABLO[16] packages in R. For differential gene expression using MAST, we modeled differential gene expression using patient ids as a random effect and resulting *p*-values were corrected for multiple testing using the BH method. For gene set enrichment analysis using fgsea, we used the fgseaMultilevel function and adjusted *p*-values using the BH method. For receptor-ligand analysis using CellChat, Wilcoxon signed-rank tests were used, and *p*-values were adjusted using the BH method. For further details on their implementation and parameters, see respective sections for each package in the methods.

**Reporting summary**

Further information on research design is available in the Nature Portfolio Reporting Summary linked to this article.

## Data availability

The data files used to produce the results reported in this article are available on Gene Expression Omnibus (GEO), dbGaP or Dryad. The computable matrix of the plasma cytokine data is deposited at Dryad [https://doi.org/10.7272/Q6MS3R18]. The raw and processed sequencing data for COMET samples used here is available at GEO under GSE237180 SuperSeries. The FASTQ files and processed data files for the bulk RNA-seq data are available at GEO (GSE237109) [https://www.ncbi.nlm.nih.gov/geo/query/acc.cgi], dbGaP (phs002686.v1.p1) and at ImmPort (SDY1760). The cellranger-processed raw feature-barcode matrices for tracheal aspirate and whole-blood are available at GEO (GSE236030) [https://www.ncbi.nlm.nih.gov/geo/query/acc.cgi], and the associated raw FASTQ files for 10X libraries have been deposited in the Sequence Read Archive (SRA) https://www.ncbi.nlm.nih.gov/Traces/study/?acc=PRJNA988459. A subset of the whole-blood data published in our previous article[10] was obtained from GSE163668 (HS1 and HS2 from GSM4995425, HS50 from GSM4995430, and the healthy controls from GSM4995449- GSM4995462) The whole-blood data reported in Sinha et al. was secured from GSE157789 and the BAL data in Liao et al. from GSE145926. The accession numbers and sample metadata are included in Supplementary Data File 2. Source data are provided with this paper.

## Code availability

Code used to generate the analysis results are available at https://github.com/UCSF-DSCOLAB/COVID-dex.

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

## Acknowledgements

This project was funded in part by the National Institutes of Health (U19AI077439, supporting the UCSF component of the NIAID Immuno-phenotyping Assessment in a COVID-19 Cohort [IMPACC] Network) and in part by Genentech (TSK-020586). We would like to thank the full COMET and IMPACC network consortia for their support and feedback in this work. LN and CSC are supported by R35HL140026. A.S. was supported by F32HL151117 and K23HL163491. GF is supported by U01DE028891-01A1, R01AI093615-11, R01DK103735, P30AR070155-05, U01AI168390, R01AI170239, P30 AI027763-31, R01DE032033, and support from the Bill and Melinda Gates Foundation and Eli Lily. GF and RP are additionally supported by the UCSF Bakar ImmunoX Initiative. We would like to thank Dr. Adam Olshen, Professor of Biostatistics, for his help in reviewing the statistical methods.

## Author contributions

L.P.A.N., R.K.P., A.S. and G.K.F. conceived of, designed, led, and executed the study and wrote the manuscript. L.P.A.N., R.K.P. and A.S. performed the data analysis. A.W. and S.C.H. provided critical support in data and study management. K.N.K. and C.M.H. provided advice on the selection of patients. W.L.E. generated the bulk RNA-seq dataset and provided additional advising. D.J.E., M.F.K., C.M.H., P.G.W., C.R.L. and C.S.C. led the COMET study and provided advice on the analysis and edits to the manuscript. The COMET Consortium performed the COMET study including patient recruitment, sample collection, data generation, and data management.

## Competing interests

The authors declare no competing interests.

## Additional information

## UCSF COMET Consortium

K. Mark Ansel[11,12], Stephanie Christenson[1], Michael Adkisson[2], Walter Eckalbar[2,4], Lenka Maliskova[2], Andrew Schroeder[2], Raymund Bueno[10], Gracie Gordon[13,14,15], George Hartoularos[15], Divya Kushnoor[6], David Lee[10], Elizabeth McCarthy[7,11,16], Anton Ogorodnikov[11], Matthew Spitzer[11,17], Kamir Hiam[7,11,16,17,18], Yun S. Song[7,19,20], Yang Sun[10], Erden Tumurbaatar[10], Monique van der Wijst[10,14,21], Alexander Whatley[19], Chayse Jones[1], Saharai Caldera[8], Catherine DeVoe[8], Paula Hayakawa Serpa[8], Christina Love[8], Eran Mick[7,8], Maira Phelps[7], Alexandra Tsitsiklis[8], Carolyn Leroux[1], Sadeed Rashid[16], Nicklaus Rodriguez[16], Kevin Tang[16], Luz Torres Altamirano[16], Aleksandra Leligdowicz[22], Michael Matthay[4,9,23], Michael Wilson[24], Jimmie Ye[10,14,25,26], Suzanna Chak[1,22], Rajani Ghale[1], Alejandra Jauregui[1], Deanna Lee[1,23], Viet Nguyen[1,23], Austin Sigman[1], Kirsten N. Kangelaris[22], Saurabh Asthana[2,6,27], Zachary Collins[2,6,27], Ravi Patel[2], Arjun Rao[2,6,27], Bushra Samad[2,6,27], Cole Shaw[2,27], Andrew Willmore[1], Tasha Lea[6], Gabriela K. Fragiadakis[2,10], Carolyn S. Calfee[1,4,9], David J. Erle[1,2,4,5], Carolyn M. Hendrickson[1], Matthew F. Krummel[6], Charles R. Langelier[7,8], Prescott G. Woodruff[1], Sidney C. Haller[1], Alyssa Ward[10], Norman Jones[28], Jeff Milush[28], Vincent Chan[11], Nayvin Chew[6,27,29,30], Alexis Combes[6,27,29,30], Tristan Courau[6,27,29,30], Kenneth Hu[6], Billy Huang[31], Nitasha Kumar[28], Salman Mahboob[31], Priscila Muñoz-Sandoval[11,12], Randy Parada[31], Gabriella Reeder[27,30,32], Alan Shen[6,27,29,30], Jessica Tsui[6,27,29,30], Shoshana Zha[1] & Wandi S. Zhu[11,12]

[11]Department of Microbiology and Immunology, University of California, San Francisco, CA, USA. [12]Sandler Asthma Basic Research Center, University of California, San Francisco, CA, USA. [13]Department of Bioengineering and Therapeutic Sciences, University of California, San Francisco, San Francisco, CA, USA. [14]Institute for Human Genetics, University of California, San Francisco, San Francisco, CA, USA. [15]Biological and Medical Informatics Graduate Program, University of California, San Francisco, San Francisco, CA, USA. [16]Helen Diller Family Comprehensive Cancer Center, University of California, San

Francisco, CA, USA. [17]Department of Otolaryngology, University of California, San Francisco, San Francisco, CA, USA. [18]Parker Institute for Cancer Immunotherapy, San Francisco, CA, USA. [19]Department of Electrical Engineering and Computer Sciences, University of California, Berkeley, Berkeley, CA, USA. [20]Department of Statistics, University of California, Berkeley, Berkeley, CA, USA. [21]Department of Genetics, University of Groningen, University Medical Center Groningen, 9713AV Groningen, Netherlands. [22]Department of Medicine, University of Toronto, Toronto, ON, Canada. [23]Cardiovascular Research Institute, University of California, San Francisco, San Francisco, CA, USA. [24]Weill Institute for Neurosciences, Department of Neurology, University of California, San Francisco, CA, USA. [25]Department of Epidemiology and Biostatistics, University of California, San Francisco, CA, USA. [26]Institute of Computational Health Sciences, University of California, San Francisco, CA, USA. [27]Bakar ImmunoX Initiative, University of California, San Francisco, CA, USA. [28]Core Immunology Laboratory, Division of Experimental Medicine, University of California, San Francisco, CA, USA. [29]Department of Anatomy, University of California, San Francisco, CA, USA. [30]Disease 2 Biology CoLab, University of California, San Francisco, CA, USA. [31]Department of Orofacial Sciences, School of Dentistry, University of California, San Francisco, CA, USA. [32]Biomedical Sciences Graduate Program, University of California, San Francisco, CA, USA.

