## [Peer Review File · Nature Communications]

Distinct pulmonary and systemic effects of dexamethasone in severe COVID-19REVIEWER COMMENTS

Reviewer #1 (Remarks to the Author):

This is an interesting article that assesses scRNA-Seq and serum proteome analysis from COVID-19 samples in the pre- and post-dexamethasone eras.

The paper is very well written.

The novelty of the work is well placed into the context of the prior literature.

The limitations of the work, primarily in relation to sample size, is also stated.

The statistical analysis seems very strong.

The use of external datasets is valuable.

Analyses such as this are very important reference atlases for the scientific literature. However, the challenge is that they often do not have clear messages for the individual reader. There is a sense of this in this report with several summary statements such as "We observed several cell- and compartment-specific differences in gene expression associated with dexamethasone treatment, emphasizing the importance of studying respiratory illness biology not only systemically, but also at the site of injury". But little biological summary of what may be occurring.

For instance the identification of steroid-resistant ISG pathways seems of interest, given therapeutic observations, and could be a potential point for the abstract.

Similarly the work seems to suggest a reduction in T cell activation after dexamethasone, but what this means for pathophysiology or viral load is not discussed.

Some more biological summary of the pathways in the discussion would help the reader with a 'message'. I appreciate that it may be difficult to 'over simplify' the datasets but it could be considered.

Reviewer #2 (Remarks to the Author):

The topic is relevant and interesting although the field is saturated with COVID-19 work. While dexamethasone has been studied before, the focus on distinct pulmonary and systemic effects appears to be relatively novel.

The authors utilized both bulk and single-cell RNA sequencing techniques on samples from the lower respiratory tract and blood, in addition to conducting plasma cytokine profiling. Their investigation focused on assessing the impact of dexamethasone on systemic and pulmonary immune cells. Their findings revealed that dexamethasone treatment led to a reduction in markers associated with antigen presentation, T-cell recruitment, and viral-induced damage in patients.

Here are my comments:

1) The description of "signatures of antigen presentation, T-cell recruitment, and viral injury" is somewhat broad and leaves room for interpretation. Could you provide more specific insights that would be valuable for shaping future research or deepening our understanding of the treatment? I believe highlighting a specific novel insight at the gene level in the abstract could be more informative than discussing broad functional categories. This approach might offer readers a clearer understanding of the study's unique contributions.

2) What is not clear in the manuscript is which data is entirely novel and what constitutes a re-analysis of previously published work. My concern is to discern what is being uniquely presented here that was not covered in the original publications. The methods section gives the impression that all the data was generated for this study. If certain methods (e.g., library preparation and sequencing) were described in the original publication, the authors should provide the accession ID and clarify that the dataset was reused. Failing to do so leaves the reader with the mistaken impression that significantly more resources and effort were invested in generating new data for this paper. It would be nice to have a figure (could be supp) describing the studies/datasets/cohorts utilized in the manuscript and where did they come from.

3) Fig 1.D: The mention of 4,000 differentially expressed genes (DEGs) seems quite extensive. With such a large list, it becomes relatively straightforward to identify interesting genes. Did the authors observe IL6, IFN γ , and IL10 among the DEGs? In essence, do the transcriptomic changes align with the observed cytokine alterations?

4) The methods section states that "Gene set enrichment analysis was performed with the fgsea package (v.14.0) and the REACTOME gene set database." While the tools used for GSEA analysis are mentioned, the methodology could benefit from further detail. Specifically, did the authors use shrunk log₂ fold-change values for ranking? And if so, did they consider only the DEGs or include all genes in the analysis?

NCOMMS-23-34701-T

Response to Reviewers

REVIEWER COMMENTS – in blue (author responses is black)

Reviewer #1 (Remarks to the Author):

This is an interesting article that assesses scRNA-Seq and serum proteome analysis from COVID-19 samples in the pre- and post-dexamethasone eras.

The paper is very well written.

The novelty of the work is well placed into the context of the prior literature.

The limitations of the work, primarily in relation to sample size, is also stated.

The statistical analysis seems very strong.

The use of external datasets is valuable.

Analyses such as this are very important reference atlases for the scientific literature. However, the challenge is that they often do not have clear messages for the individual reader. There is a sense of this in this report with several summary statements such as "We observed several cell- and compartment-specific differences in gene expression associated with dexamethasone treatment, emphasizing the importance of studying respiratory illness biology not only systemically, but also at the site of injury". But little biological summary of what may be occurring.

For instance the identification of steroid-resistant ISG pathways seems of interest, given therapeutic observations, and could be a potential point for the abstract.

Similarly the work seems to suggest a reduction in T cell activation after dexamethasone, but what this means for pathophysiology or viral load is not discussed.

Some more biological summary of the pathways in the discussion would help the reader with a 'message'. I appreciate that it may be difficult to 'over simplify' the datasets but it could be considered.

We thank the reviewer for the kind perspective on the novelty, our prose, and statistical and analytical approaches taken in our work.

We agree that presenting some clear messages to the reader of what may be occurring biologically is important. As the reviewer points out, some of the most salient observations are the signatures of decreased antigen presentation and leukocyte recruitment and activation, as well as the notable increase in some ISGs. Our findings suggest a possible mechanism of dexamethasone's therapeutic benefit, potentially that the decreased antigen presentation and

recruitment signify a decrease in immune viral clearance, which is consistent with observations that viral antigen is increased in steroid-treated patients and that viral load is higher in steroid-treated animal models.

We have revised the work in the following ways to clarify these messages to the reader:

- In the **discussion** we have added several statements that make these findings more explicit. In addition, we add to the discussion a paragraph discussing the therapeutic implications of our findings, both as how we can further interpret results of clinical trials such as in the RECOVERY trial as well as further targets to consider such as selectin-P ligand to better refine our treatments for severe pulmonary illness.
- In the **abstract** we have added specific findings including the compartment specific steroid-resistant ISG pathway, as well as the MHC II and selectin-P ligand signaling, and T cell recruitment via ICAM and integrin signaling.

We thank the reviewer for these comments and feel this very much strengthens and clarifies our message to readers.

Reviewer #2 (Remarks to the Author):

The topic is relevant and interesting although the field is saturated with COVID-19 work. While dexamethasone has been studied before, the focus on distinct pulmonary and systemic effects appears to be relatively novel.

The authors utilized both bulk and single-cell RNA sequencing techniques on samples from the lower respiratory tract and blood, in addition to conducting plasma cytokine profiling. Their investigation focused on assessing the impact of dexamethasone on systemic and pulmonary immune cells. Their findings revealed that dexamethasone treatment led to a reduction in markers associated with antigen presentation, T-cell recruitment, and viral-induced damage in patients.

Here are my comments:

1) The description of "signatures of antigen presentation, T-cell recruitment, and viral injury" is somewhat broad and leaves room for interpretation. Could you provide more specific insights that would be valuable for shaping future research or deepening our understanding of the treatment? I believe highlighting a specific novel insight at the gene level in the abstract could be more informative than discussing broad functional categories. This approach might offer readers a clearer understanding of the study's unique contributions.

We thank the reviewer for this review and for recognizing the novelty of our analysis of the compartment specific effects of dexamethasone in the context of severe COVID-19.

We agree that adding specificity to our observations and their implications for understanding dexamethasone treatment would strengthen our manuscript. To this end we have added the following:

- In the **abstract** we have added specific insights including the compartment specific steroid-resistant ISG pathway, as well as the MHC II and selectin-P ligand signaling, and T cell recruitment via ICAM and integrin signaling.
- In the **discussion** we have additionally added statements that summarize our findings and conclusions, as well as an additional paragraph that discusses the therapeutic implications of our findings and how our results could be valuable to shaping future studies and potential treatment, including the targeting of selectin-P ligand.

2) What is not clear in the manuscript is which data is entirely novel and what constitutes a re-analysis of previously published work. My concern is to discern what is being uniquely presented here that was not covered in the original publications. The methods section gives the impression that all the data was generated for this study. If certain methods (e.g., library preparation and sequencing) were described in the original publication, the authors should provide the accession ID and clarify that the dataset was reused. Failing to do so leaves the reader with the mistaken impression that significantly more resources and effort were invested in generating new data for this paper. It would be nice to have a figure (could be supp) describing the studies/datasets/cohorts utilized in the manuscript and where did they come from.

We apologize for any confusion in regards to which components of the data are presented for the first time in this work as compared to what data has been presented in other publications. This study reflects a combination of the two—we have updated the methods text to clarify this for each modality and sample set. To further clarify this, we have updated **Supplementary File 2** to include additional notes per entry to make sure this is fully clear to the reader on a per-sample basis. Per the reviewer's suggestion, we added an **Extended Figure 8** (copied below) that plots where data of each modality comes from (i.e. the current study v. which specific prior publication).

Of note, while some of this data has been included in prior publications, the examination of the effect of dexamethasone is uniquely presented here.

3) Fig 1.D: The mention of 4,000 differentially expressed genes (DEGs) seems quite extensive. With such a large list, it becomes relatively straightforward to identify interesting genes. Did the authors observe IL6, IFN γ , and IL10 among the DEGs? In essence, do the transcriptomic changes align with the observed cytokine alterations?

While we do not necessarily expect alignment of bulk blood transcriptomics and circulating plasma biomarker levels, we did investigate this question. Notably, blood gene expression levels of IL-6 and IFN-gamma were not significantly different between dexamethasone-treated and non-dexamethasone-treated patients. IL-10 gene expression however, was found to be significantly higher in dexamethasone-treated patients, which aligns with the difference observed in plasma biomarker levels.

4) The methods section states that "Gene set enrichment analysis was performed with the fgsea package (v.14.0) and the REACTOME gene set database." While the tools used for GSEA analysis are mentioned, the methodology could benefit from further detail. Specifically, did the authors use shrunk log₂ fold-change values for ranking? And if so, did they consider only the DEGs or include all genes in the analysis?

We agree with the reviewer that these specifics of the methodology are important for the readership to understand and reproduce our findings. For all gene set enrichment analyses, we used the full list of genes in the analysis, and ranked genes by shrunken log fold change. The corresponding bulk and single cell analysis sections in the Methods were updated to include those additional details (yellow highlights).

REVIEWER COMMENTS

Reviewer #1 (Remarks to the Author):

The authors have responded well to the Reviewer's comments.

Reviewer #2 (Remarks to the Author):

I think that authors have satisfactorily addressed all my concerns and suggestions. Their efforts to incorporate specific insights (at gene level!) into antigen presentation, T-cell recruitment, and the impact of viral injury, as well as to clarify the novel aspects versus re-analyzed data in your study, have enhanced the clarity and depth of your manuscript. However, I still miss the name of key genes in abstract. The additional methodological details provided for the gene set enrichment analysis and the alignment of transcriptomic changes with cytokine alterations were important. Finally, the effort to delineate new data from re-analyzed data add substantial value to the manuscript. The manuscript has been improved a lot.

Reviewer #3 (Remarks to the Author):

Naturecommunications-444663

The statistics section is very scratch to non-existent. It is not clear which test was used to calculate the p-value prior to the adjustment. The named adjustment is relatively simple and it may be better to include it in the text. P-values are mentioned through out the paper/figures with "adj." or "non adj." . It would be better to make them consistent or specify the test that produced the p-values (very likely from different methods).

Page 7, line 205, what is the "concordance" measure and what is "R"? It seems to me that three packages "Diablo", "MAST" and "CellChat" were used for data analysis. I would suggest the authors to find a statistician who is familiar with these packages (or willing to check out the contents in the packages) as a co-author to rewrite the statistics section to clearly state the methods within these packages for a better interpretation of the results.

UCSF has a department of "epidemiology and biostatistics". There should be someone who can add a valid a statistics section to the manuscript. You may try Dr. Fei Jiang: fei.jiang@ucsf.edu first.

REVIEWER COMMENTS AND RESPONSES

- Reviewer comments in black
- Author responses in blue
- Changes to the text in revised manuscript highlighted in yellow

Reviewer #1 (Remarks to the Author):

The authors have responded well to the Reviewer's comments.

Thank you for this follow up—we are glad we were able to successfully address the reviewer's comments.

Reviewer #2 (Remarks to the Author):

I think that authors have satisfactorily addressed all my concerns and suggestions. Their efforts to incorporate specific insights (at gene level!) into antigen presentation, T-cell recruitment, and the impact of viral injury, as well as to clarify the novel aspects versus re-analyzed data in your study, have enhanced the clarity and depth of your manuscript. However, I still miss the name of key genes in abstract. The additional methodological details provided for the gene set enrichment analysis and the alignment of transcriptomic changes with cytokine alterations were important. Finally, the effort to delineate new data from re-analyzed data add substantial value to the manuscript. The manuscript has been improved a lot.

We thank the reviewer for this follow up and are pleased that our updates to the manuscript significantly improved the work. We have revised our abstract to highlight genes suggesting decreased T cell activation including TNFSFR4 and IL1R.

Reviewer #3 (Remarks to the Author):

Naturecommunications-444663

The statistics section is very scratch to non-existent. It is not clear which test was used to calculate the p-value prior to the adjustment. The named adjustment is relatively simple and it may be better to include it in the text. P-values are mentioned throughout the paper/figures with “adj.” or “non adj.” . It would be better to make them consistent or specify the test that produced the p-values (very likely from different methods).

We thank the reviewer for identifying areas where statistical methods can be clarified or made more explicit. The adjustments are Benjamini-Hochberg (BH) and we now have specified this in each place we report an adjusted p-value and include that in our Statistics section of the methods. In each case we additionally report the test used.

Page 7, line 205, what is the “concordance” measure and what is “R”? It seems to me that three packages “Diablo”, “MAST” and “CellChat” were used for data analysis. I would suggest the authors to find a statistician who is familiar with these packages (or willing to check out the contents in the packages) as a co-author to rewrite the statistics section to clearly state the methods within these packages for a better interpretation of the results.

The R is a Spearman’s correlation rho, which has been added to the text and can also be found in the Figure 3 legend. The three packages that the reviewer mentions (DIABLO, MAST, and CellChat) each have a section in the methods that describe their implementation, which have now been extended with further details on their statistics. For clarity we have now additionally mentioned them in our Statistics section with statistical tests, and a reference to their specific sections. Should the editors prefer these sections moved to the statistics section based on journal style, we are happy to have them moved there.

UCSF has a department of “epidemiology and biostatistics”. There should be someone who can add a valid a statistics section to the manuscript. You may try Dr. Fei Jiang: fei.jiang@ucsf.edu first.

We consulted Dr. Adam Olshen, Professor of Biostatistics at UCSF, who reviewed methods and helped revise our statistical methods sections for clarity and substantial additional detail and now finds each to be appropriate and satisfactory. Based on the level of input he needed to provide, he felt it was more appropriate to be included in the acknowledgements, rather than as a co-author. We have added a sentence thanking Dr. Olshen for his suggestions to the acknowledgements section.

REVIEWERS' COMMENTS

Reviewer #3 (Remarks to the Author):

I am fine with the added statistical section and the authors' responses.